# Effects of Different Yeasts on Physicochemical and Oenological Properties of Red Dragon Fruit Wine Fermented with *Saccharomyces cerevisiae*, *Torulaspora delbrueckii* and *Lachancea thermotolerans*

**DOI:** 10.3390/microorganisms8030315

**Published:** 2020-02-25

**Authors:** Xiaohui Jiang, Yuyun Lu, Shao Quan Liu

**Affiliations:** 1Department of Food Science and Technology, National University of Singapore, Science Drive 2, Singapore 117543, Singapore; jiangxh@nus.edu.sg; 2National University of Singapore (Suzhou) Research Institute, 377 Lin Quan Street, Suzhou Industrial Park, Suzhou 215123, Jiangsu, China

**Keywords:** red dragon fruit, alcoholic fermentation, *Saccharomyces cerevisiae*, *Torulaspora delbrueckii*, *Lachancea thermotolerans*

## Abstract

A new type of fruit wine made from red dragon fruit juice was produced through alcoholic fermentation (AF) with different yeasts: *Saccharomyces cerevisiae* EC-1118, *Torulaspora delbrueckii* Biodiva and *Lachancea thermotolerans* Concerto. Complete AF with similar fermentation rates in terms of sugar utilisation and ethanol production (8–9%, *v*/*v*) was achieved by three yeast strains. *T. delbrueckii* produced a significantly lower amount of glycerol and acetic acid, while *L. thermotolerans* produced more lactic and succinic acids. In addition, the two non-*Saccharomyces* strains were more efficient in proline utilisation. For volatile compounds, *S. cerevisiae* produced the highest amounts of esters, while *T. delbrueckii* produced more higher alcohols, isoamyl acetate and terpenes. On the other hand, AF caused significant degradation of betacyanin pigments and total phenolic compounds. Nevertheless, better retention of antioxidant activity and colour stability was found in *L. thermotolerans* and *T. delbrueckii* fermented wines than that of *S. cerevisiae*. This study suggested that it is feasible to use pure non-*Saccharomyces* yeast to produce red dragon fruit wine for commercialization.

## 1. Introduction

Dragon fruit is the fruit of the cactus genus *Hylocereus* that originated from central Mexico and South America. Nowadays, it is cultivated in many Southeast Asian countries as well as Australia, Israel and Reunion Island. This fruit has an ovoid shape and its skin is mostly pinkish-red in colour and covered with large red or green scales. The common edible varieties are *Hylocereus undats* (white pulp), *Hylocereus polyrhizus* and *Hylocereus costaricensis* (purple–red pulp). Dragon fruit is rich in vitamins, minerals and carbohydrates especially reducing sugars, including fructose (0.4–2.0 g/100 g pulp) and glucose (3.0–5.5 g/100 g pulp) [1]. Dragon fruit is also known for its high antioxidant capacity, which is mainly attributed to the purple–red pigment, betalain, found in the skin and pulp [2]. Besides the strong antioxidant capacity, betalains also possess anti-cancer, anti-lipidemic and antimicrobial effects [2,3].

However, dragon fruit is highly perishable and commonly consumed fresh or used for juice and puree production. The processing of dragon fruit into clear juice in industry is challenging due to its significantly high contents of mucilaginous polysaccharides (e.g., pectin) and tiny black seeds [4]. Therefore, alcoholic fermentation (AF) of dragon fruit into alcoholic beverages or fruit wine may provide an alternative for postharvest preservation and add value to the fruit.

Yeast strain selection for AF is one of the most important factors that affects wine quality and fermentation process. *Saccharomyces cerevisiae* is widely used in the modern wine industry to produce wines with reliable and predictable quality, owing particularly to its remarkable sugar consumption and ethanol production and tolerance. Non-*Saccharomyces* yeasts were previously regarded as inducing a negative impact on wine quality, while recently, some researchers believed that they could improve flavour complexity and mouthfeel through the production of a range of enzymes or metabolites [5]. *Torulaspora delbrueckii* is one of the first *non-Saccharomyces* yeasts that has been commercially produced [6]. This strain is characterised by the low production of acetaldehyde and acetic acid, but high tolerance to osmotic shock, which makes it particularly adaptable in high sugar must environment, such as ice or botrytized grapes [7]. *T. delbrueckii* can positively improve wine aroma by producing high concentrations of esters, volatile fruity thiols, monoterpenes and low amounts of higher alcohols [8,9]. *Lachancea thermotolerans* is another popular non-*Saccharomyces* species that has been investigated in fruit wine fermentation [10]. *L. thermotolerans* is characterised by its high production of lactic acid, low production of volatile acidity and undesirable odour-active compounds, as well as moderate alcohol production. It is often used in low acidity must to enhance roundness and balanced acidity.

To date, little study has been done on dragon fruit juice fermentation [11]. The aim of this study was to develop a new type of fruit wine, red dragon fruit wine, through AF of red dragon fruit juice with selected *Saccharomyces* and non-*Saccharomyces* yeast strains. The fermentation performance and impact on wine quality were investigated for red dragon fruit juice fermented with three yeast strains, *S. cerevisiae*, *T. delbrueckii* and *L. thermotolerans,* in terms of physicochemical and oenological properties, including volatile composition, antioxidant activity and colour stability.

## 2. Materials and Methods

### 2.1. Yeast Strains and Media

*Saccharomyces cerevisiae* Lalvin EC-1118 (Lallemand Inc., Montreal, QC, Canada), *Torulaspora delbrueckii* Biodiva and *Lachancea thermotolerans* Concerto (Lallemand Inc., Brooklyn Park, Australia) were obtained in freeze-dried form. The freeze-dried strains were propagated in nutrient broth comprised of 2% (*w*/*v*) glucose, 0.25% (*w*/*v*) yeast extract, 0.25% (*w*/*v*) bacteriological peptone and 0.25% (*w*/*v*) malt extract. Yeast cultures were incubated at 25 °C for 48 h and then stored at −80 °C until use.

### 2.2. Dragon Fruit Juice Preparation and Fermentation Conditions

Red dragon fruits (*Hylocereus costaricensis*) were purchased from a local wholesale market. The skins were removed, and the pulp was prepared using a domestic blender and stored at −20 °C before use. Dragon fruit juice (pH 4.7, °Brix 13.1) was adjusted to pH 4.0 and °Brix 20 with 1M DL-malic acid and sucrose, respectively. The adjusted juice was then pasteurised at 60 °C for 20 min. The effectiveness of pasteurisation was checked and verified by spreading plates on potato dextrose agar (PDA, for yeasts) and de man, rogosa and sharpe (MRS) agar (for bacteria), respectively. A starter culture was prepared by inoculating 10% (*v*/*v*) propagated pure yeast culture (in nutrient broth) into the pasteurised red dragon fruit juice at 25 °C for 72–96 h to achieve a yeast cell count of 10^7^ CFU/mL. For fermentation, a 1% (*v*/*v*) starter culture was inoculated into the pasteurised red dragon fruit juice with the initial yeast cell count of ~10^5^ CFU/mL. Triplicate fermentations of 200 mL of juice were carried out in 500 mL sterile conical flasks at 20 °C for 14 days. Sampling was done at 2 days’ interval for 14 days to monitor °Brix, pH, viable yeast cell count, sugars, ethanol, total phenolic content and betacyanin content.

### 2.3. Measurement of °Brix, pH and Viable Cell Count

°Brix and pH values of the red dragon fruit juice or wines were measured with a refractometer (ATAGO, Tokyo, Japan) and a pH meter (Metrohm, Switzerland), respectively. The viable cell count was determined by spreading plates on PDA (Oxoid, Hampshire, England).

### 2.4. Analysis of Non-Volatile Components and Ethanol with HPLC

Red dragon fruit juice or wine samples were centrifuged at 10,000× *g*, 10 °C for 10 min (Eppendorf 5810R, Hamburg, Germany) and filtered with 0.20 μm RC membrane filter (Sartorius, Gottingen, Germany) before injection. Sugars were analysed with Zorbax carbohydrate column (150 × 4.6 mm, Agilent, Santa Clara, CA, USA) in a Shimadzu HPLC system with an evaporative light scattering detector (ELSD). Isocratic elution was used with a mobile phase of acetonitrile and water (80:20, *v*/*v*) at a flow rate of 1.4 mL/min at 40 °C. A Supelcogel C-610 column (300 × 7.8 mm, Supelco, Sigma–Aldrich, Barcelona, Spain) was used to analyse organic acids, glycerol and ethanol. Organic acids were analysed in the same Shimadzu HPLC system as the sugar analysis using a photodiode array detector with 0.1% (*v*/*v*) sulphuric acid as the mobile phase at a flow rate of 0.4 mL/min at 40 °C. Glycerol and ethanol were analysed in a Water HPLC system (Waters, MA, USA) connected to a refractive index detector (Waters 2414) using 0.05% (*v*/*v*) sulphuric acid as the mobile phase at a flow rate of 1.1 mL/min at 80 °C.

Free amino acids in red dragon fruit juice were analysed with an amino acid analyser (ARACUS, MembraPure, Hennigsdorf, Germany). Standard norleucine solution was used as an internal standard. Salicylic acid (10%, 100 µL), sample (200 µL) and norleucine (100 nmol, 200 µL) were mixed in a 1.5 mL sample vial and incubated at 4 °C for 60 min. The mixture was then centrifuged at 10,000 × *g* for 5 min, and the supernatant was filtered before analysis. The separation was carried out in a lithium system column kit (MembraPure, Hennigsdorf, Germany) consisting of a pre-column and a separation column (150 × 4 mm). Eluent solutions (A–F), wash and derivatization solution containing ninhydrin were purchased from MembraPure. The eluent flow rate and reactor temperature were set at 200 μL/min and 130 °C, respectively. The separated amino compounds were detected with an LED photometer detector at 440 and 570 nm.

### 2.5. Analysis of Volatile Compounds with GC-MS/FID

Volatile compounds were analysed using a headspace solid-phase microextraction (SPME) method coupled with gas chromatography-mass spectrometer and flame ionization detector (GC-MS/FID), as reported by Lu et al. [12]. Liquid juice or wine samples were centrifuged, and the pH was adjusted to 2.5 by 1 M HCl, and a 5 mL sample was sealed in a 20-mL vial with a septum lined with polytetrafluoroethylene (PTFE). Volatile samples were extracted using an SPME autosampler (CTC, Combi Pal, Switzerland) with a Carboxen/PDMS fibre[p] (85 µm, Supelco, Sigma–Aldrich, Barcelona, Spain) at 60 °C for 50 min with 250 rpm agitation. The fibre was then thermally desorbed into the injector port of Agilent 7890A gas chromatograph (Santa Clara, CA, USA) at 250 °C for 3 min. Separation of volatiles was carried out in an Agilent DB-FFAP column (60 m × 0.25 mm ID × 0.25 µm film thickness) with helium as the carrier gas at 1.2 mL/min. The GC oven temperature was programmed at 50 °C for 5 min and then increased to 230 °C at the rate of 5 °C per min and held at 230 °C for 30 min. Volatiles were identified by comparing their MS value with Wiley MS and National Institute of Standard and Technology (NIST)libraries and verified by Linear Retention Index (LRI) values. LRI values were determined using a series of alkanes (C_5_-C_40_) on the DB-FFAP column under identical conditions.

### 2.6. Measurement of CIE Colour Parameters

Colour parameters expressed as *L^*^*, *a^*^* and *b^*^* values in CIELab colour space were measured by using a spectrophotometer (Konica Minolta CM-5, Osaka, Japan) with the D65 illuminant. *L^*^* value corresponds to the lightness of sample with a value ranging from 0 (for pure black) to 100 (for pure white); *a^*^* value corresponds to redness when positive and greenness when negative; *b^*^* value corresponds to yellowness when positive and blueness when negative. Chroma *C^*^* and hue angle *h*° were calculated using the formula *C^*^* = (*a*^*2^ + *b*^*2^)^1/2^ and *h*° = arctan (*b*^*^/*a*^*^), respectively.

### 2.7. Determination of Betacyanin Content (BC)

BC was determined using a spectrometric method, as described by Stintzing et al. [13]. After centrifugation, the absorbance of red dragon fruit wines was measured at 538 nm by using a UV spectrometer (Shimadzu UVmini -1240, Kyoto, Japan). The BC, expressed as betanin equivalents (BE), was calculated by the following equation:
(1)$$\mathrm{Betacyanin}\;(\mathrm{mg}/L)=\frac{A_{538}\times \mathrm{MW}\times \mathrm{DF}\times 1000}{\varepsilon \times L}$$
where A_538_ is the absorbance at 538 nm; DF is the dilution factor; L is the path length of cuvette (L = 1 cm); MW is the molecular weight of betanin (MW = 550 g/mol); ɛ is the molar extinction coefficient of betanin (ɛ = 60,000 L/mol*cm).

### 2.8. Total Phenolic Content (TPC) and Total Antioxidant Capacity (TAC)

Total phenolic contents were determined using a Folin–Ciocalteu method as described by Isabelile et al. [14] with a microplate reader (Multiskan Spectrum, Thermo Scientific, Milford, MA, USA). The concentration of total phenolic compounds was expressed as mg/L of gallic acid equivalents (GAE) in a fresh sample.

Total antioxidant capacity was measured by using the oxygen radical absorbance capacity (ORAC) assay [15]. Total antioxidant capacity was expressed as mM of Trolox equivalents (TE) in a fresh sample.

### 2.9. Statistical Analysis

Statistical analysis was carried out by one-way ANOVA (analysis of variance) and Tukey’s test at a 95% confidence level using SPSS software (SPSS Corporation, Chicago, IL, USA, version 17.0). Results were considered statistically significant if *p* < 0.05.

## 3. Results

### 3.1. Growth and Metabolic Characteristics of Yeast Strains

The evolution of yeast cell counts in red dragon fruit wine fermentation with *L. thermotolerans* Concerto, *T. delbrueckii* Biodiva and *S. cerevisiae* EC1118 is shown in Figure 1. The cell count of *S. cerevisiae* gradually increased from 8.12 × 10^5^ CFU/mL to 8.71 × 10^7^ CFU/mL by day 10 and maintained until the end of the fermentation (day 14). In contrast, the cell count of *T. delbrueckii* reached its maximum on day 10 (3.31 × 10^8^ CFU/mL) and gradually declined thereafter, and *L. thermotolerans* reached its maximum on day 6 (2.40 × 10^8^ CFU/mL). *S. cerevisiae* had a slower growth rate at the exponential phase compared to the two non-*Saccharomyces* strains.

All three yeast strains exhibited similar evolution patterns in terms of the changes in °Brix, fructose, glucose and ethanol during AF (Figure 2). The decrease of °Brix started from day 2 and stabilised at day 8 for *S. cerevisiae*, and day 10 for both *T. delbrueckii* and *L. thermotolerans,* corresponding to the exhaustion of sugars (fructose and glucose). The sucrose added into the juice before fermentation for the purpose of adjusting °Brix was not detected in any of the samples throughout AF. At the end of the AF, all three treatments reached similar final °Brix (8.3 to 8.6) and ethanol level (8.3–8.9%, *v*/*v*) with the total sugar residues less than 2.0 g/L (Table 1), indicating complete fermentation for different yeast strains in the red dragon fruit juice.

Glycerol was detected in significant amounts in all treatments. *S. cerevisiae* produced the highest amount of glycerol (9.60 g/L), followed by *L. thermotolerans* (8.03 g/L) and *T. delbrueckii* (6.07 g/L) (Table 1). A similar trend was observed for acetic acid and pyruvic acid production with *S. cerevisiae* being the highest producer and *T. delbrueckii,* the lowest (Table 1).

*L. thermotolerans* produced more than 2-fold lactic acid (1.8 g/L) in red dragon fruit wine than the other two strains (Table 1). *L. thermotolerans* also produced the highest amount of succinic acid (1.67 g/L), while *S. cerevisiae* produced the lowest (0.86 g/L). Malic acid decreased significantly from 11.6 g/L to 6.1–6.8 g/L in red dragon fruit wines, this was likely to be the main cause for the increase in pH observed (Table 1).

### 3.2. Changes in Nitrogen-Containing Compounds

The major amino compounds present in red dragon fruit juice are proline, arginine, hydroxyproline and glutamic acid (Table 2). The total amount of free amino compounds in red dragon fruit juice was 920.65 mg/L, while the yeast assimilable nitrogen (YAN), including ammonium and alpha amino acids (without proline and hydroxyproline), was 97.28 mg N/L. The red dragon fruit wine fermented with *L. thermotolerans* contained the highest residual YAN and total nitrogen compounds (Table 2), which suggested a lower nitrogen demand or higher nitrogen releaser of the strain.

The three yeast strains showed different patterns of utilisation of individual amino compounds. The two non-*Saccharomyces* yeasts were able to utilise most of the proline in red dragon fruit juice (>90%), while *S. cerevisiae* retained a considerable amount of proline (around 54%). A significantly higher amount of GABA was detected in *L. thermotolerans* fermented wine than in the other two wine samples. Urea was not detected in red dragon fruit juice but was produced by *T. delbrueckii* and *S. cerevisiae* (Table 2). In general, hydroxyproline cannot be metabolized by *S. cerevisiae* under normal grape winemaking conditions. The observation with dragon fruit wine was quite unusual which needs further investigation.

Branched-chain and aromatic amino acids, such as leucine, isoleucine, phenylalanine and tryptophan, were taken up by yeast and metabolised to aroma compounds, such as higher alcohols and esters. Higher utilisation of these amino acids by *T. delbrueckii* and *S. cerevisiae* corresponded to the higher levels of volatile compounds production in red dragon fruit wines (Appendix A).

### 3.3. Volatile Compounds in Red Dragon Fruit Juice and Wines

In this study, 41 main volatile compounds were identified in red dragon fruit juice and wines, including fatty acids, alcohols, esters, aldehydes, ketones, terpenoids and phenolic compounds (Appendix A). Only 14 of the identified compounds were found in red dragon fruit juice, comprising mainly green acidic aroma compounds, hexanoic acid, 1-hexanol, hexanal and octanoic acid, which made up more than 85% of the total relative peak area (RPA) of identified volatile components.

Alcohols comprised more than 90% RPA in fermented red dragon fruit wines, mainly due to the high levels of ethanol (Appendix A). Higher alcohols increased significantly in the fermented red dragon fruit wines, especially isoamyl alcohol and 2-phenylethyl alcohol, both of which impart fruity and floral aroma. The two non-*Saccharomyces* yeast strains produced considerably higher concentrations of total higher alcohols than *S. cerevisiae* with *T. delbrueckii,* the highest producer (Appendix A). *T. delbrueckii* produced the highest level of isoamyl alcohol and 2-phenylethyl alcohol.

Esters are the second dominant group of volatiles in red dragon fruit wines and were constituted of mainly ethyl esters and acetate esters (Appendix A). *S. cerevisiae* fermented wine had the highest level of total esters, in particular ethyl octanoate and ethyl decanoate. *T. delbrueckii* produced the highest level of isoamyl acetate, while *L. thermotolerans* produced the highest level of ethyl acetate.

Terpenes that have pleasant floral and fruity aromas are present in small amounts in red dragon fruit juice, composed mainly of linalool. After fermentation, the total amount of terpenes increased in *T. delbrueckii* fermented wines, while a decrease was found in the other two wines. Longifolene and *β*-myrcene were the two major terpenoids been produced.

The fatty acids (e.g., hexanoic, octanoic and nonanoic acids) reduced significantly in all red dragon fruit wines. *T. delbrueckii* fermented wine had the most decrease, while *S. cerevisiae,* the least.

### 3.4. Betacyanin Content (BC), Total Phenolic Content (TPC), Total Antioxidant Capacity (TAC) and Colour Stability in Red Dragon Fruit Wines

In red dragon fruit juice, the BC, TPC and TAC values were 181.28 BE mg/L, 387.30 GAE mg/L and 8.59 TE mmol/L, respectively (Table 3). During fermentation, both BC and TPC decreased gradually but significantly, with 22% (*L. thermotolerans*) to 29% (*T. delbrueckii*) decrease in TPC and 37% (*T. delbrueckii*) to 55% (*S. cerevisiae*) decrease in BC, while the TAC values fluctuated during fermentation (Figure 3). There are strong correlations between BC and TPC in red dragon fruit wines fermented by all the three yeast strains (r = 0.97 for *T. delbrueckii*, r = 0.87 for *L. thermotolerans* and r = 0.95 for *S. cerevisiae*).

The colours of the red dragon fruit juice and wines showed statistical differences after fermentation among different yeast strains (Table 3). The *L** value increased slightly in all red dragon fruit wines. The increase in lightness agreed with betacyanin retention, with *S. cerevisiae* fermented red dragon fruit wine being the brightest (the lowest BC); *T. delbrueckii* fermented wine being the darkest (the highest BC). The increases in *a** in all red dragon fruit wines indicated an increase in redness. Significant decreases in *b** value from positive to negative for *L. thermotolerans* and *T. delbrueckii* indicated a shift from yellowness to blueness. Increases in chroma *C** for the fermented wines suggested higher purity and brighter colour. While the shifting of *h°* from 2.22 to approximately 355 in the two non-*Saccharomyces* fermented wines indicated a shift from orange–red to purple–red hue.

## 4. Discussion

The unexpected slow growth rate of *S. cerevisiae* at the exponential phase compared to the two non-*Saccharomyces* yeast strains could be due to a deficiency of nutrients in fermented dragon fruit juices, such as nitrogen which will be discussed later. The decline in the population of *T. delbrueckii* and *L. thermotolerans* towards the end of AF might be attributed to the increased ethanol concentration in the red dragon fruit juice (Figure 2), as non-*Saccharomyces* yeasts are known to be less tolerant of ethanol [16]. Despite their different growth rate, all three yeast strains completed the AF with similar speed as indicated by the similar rate of °Brix reduction, sugar consumption and ethanol production (Figure 2). These findings suggest that the two non-*Saccharomyces* yeasts showed comparable fermentation abilities to that of *Saccharomyces* yeast in red dragon fruit wine fermentation. This observation is in good agreement with several studies, demonstrating that *T. delbrueckii* and *L. thermotolerans* are relatively powerful fermenters among non-*Saccharomyces* and when used in monoculture fermentation. The rate of sugar consumption is comparable to that of *S. cerevisiae* [17,18,19]. Glucose and fructose were largely metabolised by yeasts into ethanol. Sucrose was added into the initial red dragon fruit juice to obtain comparable °Brix as grape juice and higher final ethanol content. The fact that no sucrose was detected in any of the samples was likely due to the endogenous enzyme invertase in dragon fruit, which rapidly hydrolysed sucrose into glucose and fructose upon adding into the red dragon fruit juice. Although the presence of invertase was not reported for red dragon fruit, high invertase activities have been found in white dragon fruit (*Hylocereeus undatus*) and prickly pear (*Opuntia* sp.), both belonging to the *Cactaceae* family [13].

Besides ethanol, glycerol is one of the major by-products of glycolysis through the glyceropyruvate pathway [12,20]. In glycolysis, glucose is first transformed into fructose-1,6-bisphosphate, which is then transformed into dihydroxy-acetone phosphate (DHAP) and glyceraldehyde-3-phosphate. The glycerol is continually produced from DHAP via the reduction of DHAP by glycerol-3-phosphate dehydrogenase to form glycerol-3-phosphate, followed by the dephosphorylation of glycerol-3-phosphate by glycerol-3-phosphatase. In grape wines, the glycerol levels in most cases fall into the range of 4 to 9 g/L and are highly dependent on the style and origin [20,21]. A higher level of glycerol in wine is often associated with improvement in the overall taste balance and mouthfeel. The final concentration of glycerol found in the non-*Saccharomyces* yeast fermented red dragon fruit wine showed similar results to studies on durian wine fermentation [12,22]. *S. cerevisiae* EC-1118 has been reported to be a high glycerol producer which can accumulate up to 8–9 g/L in wine [23]. In addition, *L. thermotolerans* has also been recognised as a good glycerol producer [17,24].

Malic acid was added into the juice prior to inoculation with the purpose of adjusting the pH of the juice (from 4.66 to 4.03), which would explain the high concentration of malic acid at the beginning of fermentation. The decrease of malic acid in red dragon fruit wines fermented by *S. cerevisiae* was most likely due to a passive diffusion into yeast cells rather than being metabolised [12]. While both *T. delbrueckii* and *L. thermotolerans* have been reported to be moderate malic acid consumers (20–25%), although this activity is highly strain-dependent [25].

Acetic acid is produced from the oxidation of acetaldehyde by aldehyde dehydrogenase or hydrolysis of acetyl CoA derived from pyruvate. A too high concentration of acetic acid is detrimental to the organoleptic property of wine. It was reported that the production of acetic acid and glycerol is highly correlated [26]. The relatively high acetic acid produced by *S. cerevisiae* together with glycerol could be due to certain amino acid or vitamin deficiency [27]. Lower production of acetic acid is a major characteristic of *T. delbrueckii* that has been exploited in high sugar must [7]. High production of lactic acid is one of the most important characteristics of *L. thermotolerans* [10]. A high level of lactic acid is preferred which would improve complexity to wine taste, enhance roundness and balanced acidity. Succinic acid is produced from the TCA cycle but could also result from the metabolism of γ-aminobutyric acid (GABA) [28]. A higher level of GABA in *L. thermotolerans* fermented wine may have contributed to the higher level of succinic acid produced. Succinic acid accumulates in wine at the level of 0.5–1.5 g/L; high concentrations of succinate in wine will cause a salty and bitter taste.

The quantity and variety of the nitrogen source have a pronounced effect on yeast fermentation performance and aroma development during winemaking. Generally, the minimum YAN concentration in the juice for risk-free of slow or stuck fermentation was around 140 mg N/L [29]. This indicated that the red dragon fruit is a nitrogen-deficient medium for yeast fermentation. Despite the limited nitrogen concentration, the AF was completed in red dragon fruit wine fermentation. High inoculations of pre-cultures may have contributed to fermentation performance [30]. Similar to current findings, Ciani et al. [16] observed a higher level of final YAN in mixed yeast fermentation involving *L. thermotolerans* compared to pure *S. cerevisiae* fermentation. On the other hand, the red dragon fruit wine fermented with *S. cerevisiae* had the lowest residual YAN, which indicated a higher nitrogen demand, and this might explain the slow growth rate at the exponential phase (Figure 1).

Proline is not a preferred nitrogen source for yeast, and very little would be taken up by yeast under normal fermentation conditions [31]. The high utilisation efficiency of proline by *T. delbrueckii* and *L. thermotolerans* indicated their higher preference for proline. Therefore, the high proline content in red dragon fruit juice may make it a nitrogen-adequate medium for the growth of these two non-*Saccharomyces* strains. Arginine is the major precursor of urea in grape must [32]. The higher utilisation rate of arginine by *T. delbrueckii* and *S. cerevisiae* could contribute to their urea production via yeast arginase activities (Table 2). Yeast GABA is formed through the decarboxylation of glutamate, and the catabolism of GABA results in the formation of succinate via succinic semialdehyde [28]. The high accumulation of GABA in *L. thermotolerans* fermented red dragon fruit wine could contribute to its high production of succinic acid.

For volatile profiles, only the relative abundance of volatile compounds (expressed as % relative peak area) was measured and compared among the three treatments as this study serves as a preliminary evaluation to select yeast strain(s) for red dragon fruit wine fermentation. Quantification of major volatile groups will be carried out in the future on fermentation with selected yeast strain(s), and the resulting red dragon fruit wine shall be compared with grape wine.

Upon fermentation, significant increases in the total amount of volatile components (more than a 30-fold increase in total peak area), particularly alcohols and esters, were found in all the fermented wine samples; fatty acids and aldehydes were largely catabolised. Ethanol is the most abundant volatile in fermented red dragon fruit wines. At the current ethanol level (<9%, *v*/*v*), the red dragon fruit wines were considered as low alcohol wines. *T. delbrueckii* produced a particularly high amount of higher alcohols, especially isoamyl alcohol and 2-phenylethyl alcohol which is correlated with its high utilization rate of the precursor amino acids, leucine and phenylalanine, respectively. The increase in higher alcohols production compared to *S. cerevisiae* has been reported for both *T. delbrueckii* and *L. thermotolerans* in either monoculture fermentation or mixed culture fermentation [8,24]. However, the low production of higher alcohols by these two strains has also been reported [9,17]. This discrepancy may be explained by the complexity and regulation of the Ehrlich pathway. Furthermore, the low nitrogen content in red dragon fruit juice may also lead to the high production of higher alcohols [33].

Esters belong to the most abundant group of aroma compounds found in wine, and they are mainly produced by yeast through lipid and acetyl-CoA metabolism. In monoculture fermentation, the non-*Saccharomyces* strains used in this study have been reported to produce lower levels of ethyl esters and acetate esters than *Saccharomyces* [12,34], which is consistent with our findings.

Increases in terpenes, such as α-terpineol and linalool, have been reported for mixed cultures fermentation involving *T. delbrueckii*, which had positive effects on the quality of wines made from terpenic varieties [35]. The reduction in fatty acids corresponded with the production of the respective esters. The lower levels of fatty acid detected in *T. delbrueckii* and *L. thermotolerans* are in good agreement with previous studies [8,34].

The BC, TPC and TAC of the red dragon fruit wines were evaluated to provide a better understanding of the effect of fermentation with different yeast strains. The measured values in unfermented juice (Table 3) were lower than that of previous studies [2,3,13], which could be due to different genotype, degree of maturity of fruits, extraction methods used, as well as the heat treatment [36].

In red dragon fruit juice, the majority of the betacyanins exist in their glucosylated forms, such as betanin or phyllocactin [37]. The decrease in BC could be due to the β-glucosidase activity of the yeasts that caused the hydrolysis of the glycosidic bonds in the glucosylated betacyanins and released the less stable betanidin [38].

The final values of TPC and TAC in the red dragon fruit wines (Table 3) were lower than most red wines, but similar to or higher than many white wines [39,40]. Studies have also shown that most berry and fruit wines (e.g., apple, cranberry and red currant) in general contain lower amounts of phenolic compounds than red grape wines [39]. Significant decreases in TPC and anthocyanins (29% to 90%) have been reported in fermented strawberry, pomegranate and chokeberry wines [41,42,43]. These differences in TPC and TAC among different types of fruit wines could be attributed to the difference in their biochemical compositions. The bioactive components in red dragon fruit juice comprise mainly betacyanins, flavonoids (e.g., quercetin) and phenolic acids (e.g., gallic acid and caffeic acid), in which betacyanins contribute to the highest antioxidant capacity [3].

The strong correlation between BC and TPC found in red dragon fruit wine fermented by all the three yeast strains could be attributed to the phenol structure present in betacyanins that contributed to TPC, as well as possible synergistic interactions between betanins and phenolic compounds [44]. In contrast to studies on grape wines, there is no correlation between TAC and TPC or BC during red dragon fruit wine fermentation. Similar observations have also been reported for other berry and fruit wines, especially those with high amounts of TPC [39,45]. The poor correlation between TPC and TAC for non-grape fruit wines could be explained by the difference in the composition of phenolic compounds in the raw materials and the variations in their responses to TPC/TAC assays. Further compositional analysis of the individual phenolic compound present in red dragon fruit juice/wines and their antioxidant capacity need to be carried out to clarify their effect and contribution to TAC.

According to Herbach et al. [46], the *L** value is a reliable indicator of pigment retention in betacyanin-containing solutions. This could be due to betacynins (e.g., betanin, phyllocactin and hylocerenin) in red dragon fruit juice being degraded to orange, yellow or colourless compounds (e.g., betalamic acid, *cyclo*-dopa 5-O-*β*-glucoside) upon thermal treatment. The colour results suggested a better retention of red–purple betacynin pigments in *L. thermotolerans* and *T. delbrueckii* fermented red dragon fruit wines.

## 5. Conclusions

In conclusion, all three yeast strains, *S. cerevisiae*, *T. delbrueckii* and *L. thermotolerans,* could complete AF in red dragon fruit juice and produce wine with an ethanol content of 8–9% (*v*/*v*). Different yeast strains showed similar fermentation rates but exhibited different patterns in nitrogen compound utilisation and volatile compounds production. The two non-*Saccharomyces* yeast strains (*T. delbrueckii* and *L. thermotolerans*) were more efficient in proline utilisation. *S. cerevisiae* produced more ethyl esters, while *T. delbrueckii* produced the highest level of isoamyl alcohol, 2-phenylethyl alcohol, isoamyl acetate and terpenes. In addition, the two non-Saccharomyces yeasts better retained betacyanin pigments and antioxidant capacity. The current study indicated that it is possible to use only non-*Saccharomyces* yeast to produce red dragon fruit wine as an exotic alternative to grape wine.

## Figures and Tables

**Figure 1 microorganisms-08-00315-f001:**
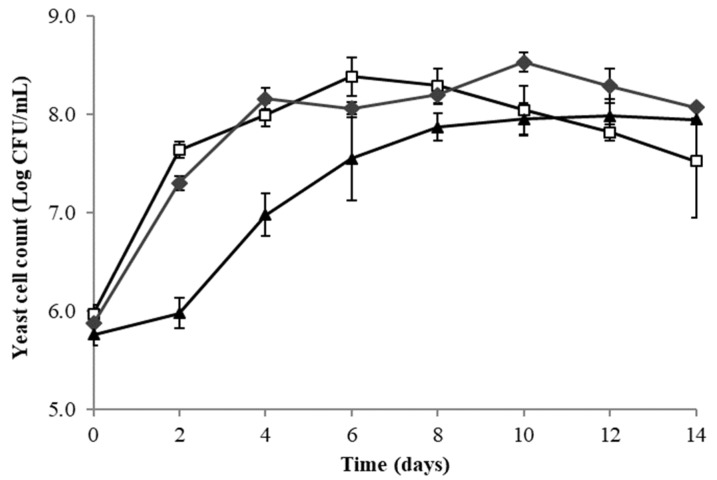
Evolution of yeast cell counts during 14-day fermentation of red dragon fruit juice by *T. delbrueckii* Biodiva (♦), *L. thermotolerans* Concerto (□) and *S. cerevisiae* EC-1118 (▲).

**Figure 2 microorganisms-08-00315-f002:**
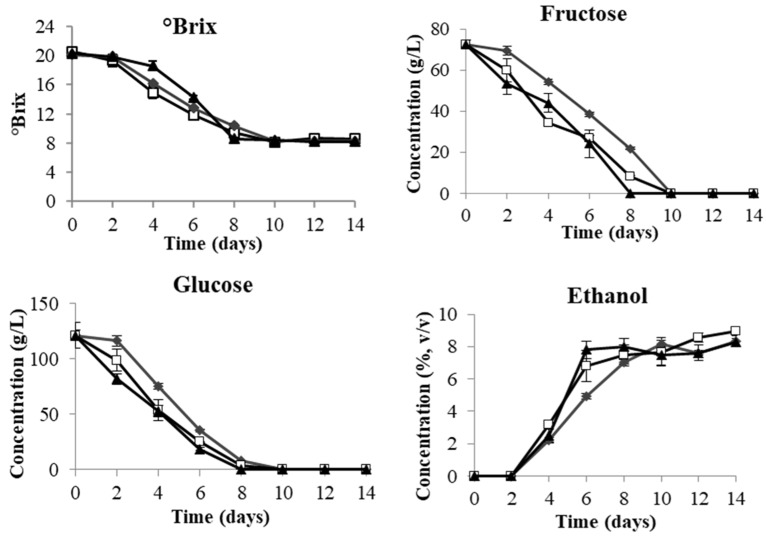
Changes in °Brix, fructose, glucose and ethanol during 14-day fermentation of red dragon fruit juice by *T. delbrueckii* Biodiva (♦), *L. thermotolerans* Concerto (□) and *S. cerevisiae* EC-1118(▲).

**Figure 3 microorganisms-08-00315-f003:**
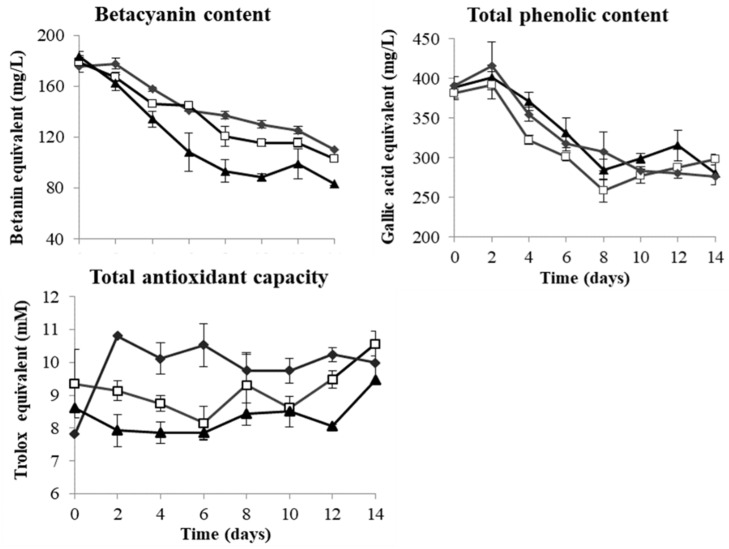
Changes in betacyanins, total phenolic content and total antioxidant capacity in red dragon fruit wine during 14-day fermentation by *T. delbrueckii* Biodiva (♦), *L. thermotolerans* Concerto (□) and *S. cerevisiae* EC-1118 (▲).

**Table 1 microorganisms-08-00315-t001:** Oenological parameters of red dragon fruit musts fermented with *Saccharomyces* and non-*Saccharomyces* yeast strains.

	Day 0	Day 14		
		Biodiva ^#^	Concerto ^#^	EC-1118 ^#^
pH	4.03 ± 0.03 ^a^	4.11 ± 0.01 ^b^	4.11 ± 0.01 ^b^	4.10 ± 0.02 ^b^
°Brix	20.33 ± 0.30 ^a^	8.25 ± 0.02 ^b^	8.60 ± 0.23 ^b^	8.24 ± 0.31 ^b^
Ethanol (%, *v*/*v*)	ND ^&^	8.33 ± 0.31 ^a^	8.93 ± 0.27 ^b^	8.25 ± 0.23 ^a^
Glycerol (g/L)	ND	6.07 ± 0.59 ^a^	8.03 ± 0.51 ^b^	9.60 ± 0.23 ^c^
Sugars (g/L)				
Fructose	71.60 ± 3.49 ^a^	ND	ND	ND
Glucose	121.18 ± 4.50 ^a^	ND	ND	ND
Organic acids (g/L)				
Acetic acid	ND	0.15 ± 0.01 ^a^	0.30 ± 0.02 ^b^	0.38 ± 0.03 ^c^
Citric acid	0.55 ± 0.01 ^a^	0.62 ± 0.03 ^b^	0.64 ± 0.04 ^b^	0.66 ± 0.06 ^b^
Lactic acid	0.87 ± 0.08 ^a^	0.75 ± 0.02 ^a^	1.80 ± 0.13 ^b^	0.83 ± 0.06 ^a^
Malic acid	11.6 ± 0.56 ^a^	6.77 ± 0.39 ^b^	6.11 ± 0.45 ^b^	6.16 ± 0.44 ^b^
Pyruvic acid	ND	0.48 ± 0.02 ^a^	0.56 ± 0.03 ^b^	0.70 ± 0.02 ^c^
Succinic acid	ND	1.27 ± 0.05 ^a^	1.67 ± 0.12 ^b^	0.86 ± 0.02 ^c^

**^#^** Biodiva, *Torulaspora delbrueckii*; Concerto, *Lachancea thermotolerans*; EC-1118, *Saccharomyces cerevisiae*. ^&^ ND: not detected. Values are given as the mean ± standard deviation (*n* = 3), and the different letters within each row are significantly different (*p* < 0.05).

**Table 2 microorganisms-08-00315-t002:** Amino compounds in red dragon fruit juice and wines fermented with *Saccharomyces* and non-*Saccharomyces* yeast strains.

Concentration (mg/L)	Day 0	Day 14		
		Biodiva ^#^	Concerto ^#^	EC-1118 ^#^
Alanine	19.36 ± 0.10 ^a^	6.55± 0.43 ^b^	14.14 ± 0.45 ^c^	2.00 ± 0.19 ^d^
Ammonium	25.01 ± 0.65 ^a^	7.50 ± 0.69 ^b^	9.47 ± 0.80 ^c^	7.70 ± 0.38 ^b^
Arginine	109.03 ± 8.63 ^a^	4.95 ± 0.23 ^b^	11.36 ± 0.56 ^c^	2.34 ± 0.17 ^d^
Asparagine	13.93 ± 0.31 ^a^	1.67 ± 0.11 ^b^	7.63 ± 0.63 ^c^	1.85 ± 0.10 ^b^
Aspartic acid	37.43 ± 0.26 ^a^	1.55 ± 0.15 ^b^	4.87 ± 0.40 ^c^	0.70 ± 0.06 ^d^
Cysteine	5.17 ± 0.49 ^a^	1.54 ± 0.03 ^b^	4.05 ± 0.16 ^c^	1.41 ± 0.024 ^d^
Glutamic acid	85.38 ± 7.60 ^a^	4.39 ± 0.10 ^b^	12.64 ± 0.45 ^c^	2.70 ± 0.10 ^d^
Glycine	3.96 ± 0.14 ^a^	2.00 ± 0.14 ^b^	4.12 ± 0.04 ^a^	1.29 ± 0.09 ^c^
Histidine	12.83 ± 0.91 ^a^	1.86 ± 0.18 ^b^	2.36 ± 0.06 ^c^	1.39 ± 0.11 ^d^
Hydroxyproline	89.64 ± 0.62 ^a^	3.31 ± 0.27 ^b^	11.03 ± 0.27 ^c^	1.70 ± 0.11 ^d^
Isoleucine	5.50 ± 0.64 ^a^	1.23 ± 0.14 ^b^	3.33 ± 0.15 ^c^	0.59 ± 0.02 ^d^
Leucine	32.37 ± 1.07 ^a^	3.98 ± 0.02 ^b^	12.99 ± 0.85 ^c^	2.26 ± 0.07 ^d^
Lysine	26.03 ± 0.19 ^a^	4.05 ± 0.19 ^b^	12.28 ± 0.22 ^c^	3.49 ± 0.33 ^b^
Methionine	9.70 ± 0.46 ^a^	1.69 ± 0.05 ^b^	3.07 ± 0.26 ^c^	1.26 ± 0.39 ^b^
Ornithine	2.42 ± 0.09 ^a^	1.55 ± 0.03 ^b^	2.63 ± 0.34 ^a^	2.37 ± 0.09 ^a^
Phenylalanine	38.58 ± 0.85 ^a^	3.64 ± 0.47 ^b^	8.80 ± 0.65 ^c^	1.59 ± 0.03 ^d^
Phosphorthanolamine	7.23 ± 0.64 ^a^	0.44 ± 0.02 ^b^	0.87 ± 0.08 ^c^	0.50 ± 0.04 ^b^
Phosphoserine	4.93 ± 0.33 ^a^	4.01 ± 0.05 ^b^	4.09 ± 0.47 ^a^	4.91 ± 0.54 ^a^
Proline	184.20 ± 5.97 ^a^	20.07 ± 1.05 ^b^	9.61 ± 0.12 ^c^	98.94 ± 4.17 ^d^
Sarcosine	52.85 ± 3.38 ^a^	6.89 ± 0.31 ^b^	18.19 ± 2.52 ^c^	8.67 ± 0.52 ^d^
Serine	12.62 ± 1.03 ^a^	1.26 ± 0.02 ^b^	4.87 ± 0.20 ^c^	1.05 ± 0.06 ^d^
Taurine	19.25 ± 1.42 ^a^	1.41 ± 0.12 ^b^	1.12 ± 0.05 ^c^	1.05 ± 0.14 ^c^
Threonine	10.92 ± 0.29 ^a^	1.20 ± 0.15 ^b^	4.37 ± 0.27 ^c^	0.65 ± 0.05 ^d^
Tryptophan	7.68 ± 0.34 ^a^	ND	3.96 ± 0.08 ^b^	ND
Tyrosine	18.05 ± 0.43 ^a^	2.96 ± 0.13 ^b^	5.90 ± 0.28 ^c^	2.09 ± 0.04 ^d^
Urea	ND ^&^	1.15 ± 0.07 ^a^	ND	3.55 ± 0.41 ^b^
Valine	11.97 ± 0.22 ^a^	1.22 ± 0.06 ^b^	4.88 ± 0.15 ^c^	ND
β-aminoisobutyric acid	49.09 ± 4.13 ^a^	44.77 ± 1.53 ^b^	52.18 ± 4.58 ^a^	36.54 ± 1.17 ^c^
ϒ-aminobutyric acid	25.52 ± 0.85 ^a^	24.44 ± 0.40 ^a^	35.40 ± 1.27 ^c^	19.81 ± 1.23 ^d^
∑ N (mg/L)	920.65	161.29	270.16	212.39
∑YAN (mg N/L)	97.28	11.16	25.85	8.76

**^#^** Biodiva, *Torulaspora delbrueckii*; Concerto, *Lachancea thermotolerans*; EC-1118, *Saccharomyces cerevisiae*. ^&^ ND: not detected. ^a,b,c,d^ Statistical analysis ANOVA (*n* = 3) at 95% confidence level with the same letter indicating no significant difference.

**Table 3 microorganisms-08-00315-t003:** Total phenolic content (TPC), total antioxidant capacity (TAC), betacyanin content (BC) and CIE Lab colour parameters of red dragon fruit juice and wines fermented with *Saccharomyces* and non-*Saccharomyces* yeast strains.

	Day 0	Day 14		
		Biodiva ^#^	Concerto ^#^	EC-1118 ^#^
TPC (mg/L GEA ^※^)	387.30 ± 4.57 ^a^	275.34 ± 5.98 ^b^	298.21 ± 6.81 ^c^	280.61 ± 14.84 ^bc^
TAC (mM TE ^※^)	8.59 ± 0.7 7 ^a^	9.99 ± 0.68 ^bc^	10.57 ± 0.38 ^b^	9.46 ± 0.74 ^c^
BC (mg/L BE ^※^)	181.28 ± 5.08 ^a^	112.24 ± 1.98 ^b^	102.90 ± 2.66 ^c^	83.18 ± 1.37 ^d^
Colour				
*L**	34.90 ± 0.25 ^a^	35.57 ± 0.05 ^b^	38.31 ± 0.17 ^c^	40.78 ± 0.85 ^d^
*a**	77.37 ± 0.04 ^a^	79.45 ± 0.06 ^b^	80.89 ± 0.23 ^c^	78.86 ± 0.16 ^d^
*b**	3.01 ± 0.64 ^a^	-8.52 ± 0.41 ^b^	−5.86 ± 0.25 ^c^	2.63 ± 0.93 ^a^
*C**	77.43 ± 0.06 ^a^	79.90 ± 0.10 ^b^	81.10 ± 0.24 ^c^	78.91 ± 0.15 ^d^
*h°*	2.22 ± 0.47 ^a^	353.9 ± 0.3 ^b^	355.9 ± 0.2 ^c^	1.53 ± 0.18 ^d^

^#^ Biodiva, *Torulaspora delbrueckii*; Concerto, *Lachancea thermotolerans*; EC-1118, *Saccharomyces cerevisiae*. ^※^ GEA, gallic acid equivalents; TE, Trolox equivalents; BE, betacyanin equivalents. ^a,b,c,d^ Statistical analysis ANOVA (*n* = 3) at 95% confidence level with the same letter indicating no significant difference.

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
