# Peer review of "Effects of Different Yeasts on Physicochemical and Oenological Properties of Red Dragon Fruit Wine Fermented with Saccharomyces cerevisiae, Torulaspora delbrueckii and Lachancea thermotolerans"

_microorganisms, 2020, doi:10.3390/microorganisms8030315_

Round 1

Reviewer 1 Report

The paper “Effects of different yeasts on physicochemical and oenological properties of red dragon fruit wine fermented with Saccharomyces cerevisiae, Torulaspora delbrueckii and Lachancea thermotolerans” by Jiang et al. is a very interesting study on the fermentation of a novel alcoholic beverage by three different yeast species. The most striking result is the ability of the non-Saccharomyces strains to fully ferment the dragon fruit juice, adding some physiological and organoleptic features, such as the efficient proline consumption and the production of interesting secondary metabolites.

The analytical data is very complete, but if the absence of human tasting by a trained panel is hard to understand in such beverage is suited for human consumption. Some metabolites as ethanol and glycerol are compared to the ones typically obtained in grape juice fermentation, but the minor metabolites (ester, higher alcohols and so on) have to be also compared to the ones usually obtained in wine fermentation. Table S1 is missing, so a full review cannot be performed.

Minor points:

Line 14. Do not used “red wine” in the abstract. You find that is not as full of color as red wines.

15, 71 and through the text: do not indicate ”var bayanus”, this is obsolete nomenclature.

Line 27, 197, 221, 262, 411, 415….name yeast species in italics.

Line 31. “central Mexico”.

Line 38. Use “betalain” in singular.

Line 52. Use full name the first time you mentioned this species, and not in line 66.

Line 81. Explain if any fermentation without pH correction has been done and why this is necessary to perform a good fermentation.

Line 177. Sucrose was used toa djust Brix.

Line 195. Separate “1.Oenological”. Change “wines” into “musts”.

Line 203. Explain the origen and metabolism of hydroxyproline, as it is used by S. cerevisiae. Is it included in the YAN?

Line 205. Indicate the YAN including proline, as the non-Saccharomyces yeasts use it.

Page 7 is again page 1.

Line 302. Explain the glyceropyruvate pathway. Glycerol is obtained from DHAP.

Line 309. The increase in pH is too little to discuss it.

Line 344. The fact that proline is osmoprotectant is not relevant here, delete.

Author Response

Responses to editor and reviewers’ comments

Reviewer 1 comments:

Comment 1: The paper “Effects of different yeasts on physicochemical and oenological properties of red dragon fruit wine fermented with Saccharomyces cerevisiae, Torulaspora delbrueckii and Lachancea thermotolerans” by Jiang et al. is a very interesting study on the fermentation of a novel alcoholic beverage by three different yeast species. The most striking result is the ability of the non-Saccharomyces strains to fully ferment the dragon fruit juice, adding some physiological and organoleptic features, such as the efficient proline consumption and the production of interesting secondary metabolites. The analytical data is very complete, but if the absence of human tasting by a trained panel is hard to understand in such beverage is suited for human consumption. Some metabolites as ethanol and glycerol are compared to the ones typically obtained in grape juice fermentation, but the minor metabolites (ester, higher alcohols and so on) have to be also compared to the ones usually obtained in wine fermentation. Table S1 is missing, so a full review cannot be performed. Need to submit Table S1.

Response: Yes, we totally agree that the sensory evaluation should be carried out to better understand this new kind of fruit wine. While, as the study is a preliminary work with the aim to develop new fruit wine through alcoholic fermentation of red dragon fruit juice with selected Saccharomyces and non-Saccharomyces yeast strains. We mainly focus on the fermentation performance of the three yeast strains (S. cerevisiae, T. delbrueckii and L. thermotolerans) in terms of physicochemical and oenological properties including volatile composition, antioxidant activity and colour stability. Sensory evaluation will definitely be conducted in our future work.

Sorry for our careless to miss the Table S1, we have added this Table in the revised manuscript in Supplementary materials.

Comment 2: Line 14. Do not used “red wine” in the abstract. You find that is not as full of color as red wines.

Response: As suggested, we have edited this in the revised manuscript.

Comment 3: 15, 71 and through the text: do not indicate “var bayanus”, this is obsolete nomenclature.

Response: As suggested, we have deleted this in the revised manuscript.

Comment 4: Line 27, 197, 221, 262, 411, 415….name yeast species in italics.

Response: As suggested, we have edited this in the revised manuscript.

Comment 5: Line 31. “central Mexico”.

Response: As suggested, we have edited this in the revised manuscript.

Comment 6: Line 38. Use “betalain” in singular.

Response: As suggested, we have edited this in the revised manuscript.

Comment 7: Line 52. Use full name the first time you mentioned this species, and not in line 66.

Response: As suggested, we have edited this in the revised manuscript.

Comment 8: Line 81. Explain if any fermentation without pH correction has been done and why this is necessary to perform a good fermentation.

Response: We did not do fermentation without pH correction. In general, the optimum fermentation pH was 3.5-4.5 for most yeasts’ fermentation (Kosseva, Joshi, & Panesar, 2016). A lower pH could help inhibit bacteria growth (prevent contamination) during fermentation and based on our previous study, pH 4.0 is the most suitable pH for Saccharomyces cerevisiae EC-1118 (Lu et al., 2017). To keep consistent among different fermentations, we adjusted the pH to 4.0.

Kosseva, M., Joshi, V. K., & Panesar, P. S. (Eds.). (2016). Science and technology of fruit wine production. Academic Press.

Lu, Y., Voon, M. K. W., Huang, D., Lee, P. R., & Liu, S. Q. (2017). Combined effects of fermentation temperature and pH on kinetic changes of chemical constituents of durian wine fermented with Saccharomyces cerevisiae. Applied Microbiology and Biotechnology, 101(7), 3005-3014.

Comment 9: Line 177. Sucrose was used to adjust Brix.

Response: As suggested, we have edited this in the revised manuscript.

Comment 10: Line 195. Separate “1. Oenological”. Change “wines” into “musts”.

Response: As suggested, we have edited this in the revised manuscript.

Comment 11: Line 203. Explain the origin and metabolism of hydroxyproline, as it is used by S. cerevisiae. Is it included in the YAN?

Response: Hydroxyproline is not included in the YAN. In general, hydroxyproline cannot be metabolized by S. cerevisiae under normal grape wine making condition (Ingledew, Magnus, & Sosulski, 1987). The observation with dragon fruit wine was quite unusual which needs further investigation.

Ingledew, W. M., Magnus, C. A., & Sosulski, F. W. (1987). Influence of oxygen on proline utilization during the wine fermentation. American Journal of Enology and Viticulture, 38(3), 246-248.

Comment 12: Line 205. Indicate the YAN including proline, as the non-Saccharomyces yeasts use it.

Response: YAN is based on nitrogen utilization of S. cerevisiae, the principle yeast used for fermentation. Since secondary amino acids such as proline or hydroxyproline are found not be metabolized by S. cerevisiae to great extend under normal wine making condition (Ingledew, Magnus, & Sosulski, 1987), they are normally categorized as yeast-non-assimilable nitrogen (YNAN) together with larger molecule weight peptides and proteins. This definition did not take into account of non-Saccharomyces yeast or when under nitrogen deficiency.

Ingledew, W. M., Magnus, C. A., & Sosulski, F. W. (1987). Influence of oxygen on proline utilization during the wine fermentation. American Journal of Enology and Viticulture, 38(3), 246-248.

Comment 13: Page 7 is again page 1.

Response: As suggested, we have edited this in the revised manuscript.

Comment 14: Line 302. Explain the glyceropyruvate pathway. Glycerol is obtained from DHAP.

Response: As suggested, we have explained the glyceropyruvate pathway in the revised manuscript. In glycolysis, glucose is firstly transformed to fructose-1,6-bisphosphate, which is then transformed to dihydroxy-acetone phosphate (DHAP) and glyceraldehyde-3-phosphate. The glycerol is continuedly produced from DHAP via the reduction of DHAP by glycerol-3-phosphate dehydrogenase to form glycerol-3-phosphate, followed by the dephosphorylation of glycerol-3-phosphate by glycerol-3-phosphatase.

Comment 15: Line 309. The increase in pH is too little to discuss it.

Response: As suggested, we have deleted the sentence in the revised manuscript.

Comment 16: Line 344. The fact that proline is osmoprotectant is not relevant here, delete.

Response: As suggested, we have deleted it in the revised manuscript.

Reviewer 2 Report

Dear authors, I have found the experimental design simple yet well conducted as to show metabolic/organoleptic differences in parameters in red dragon fruit wine. I also found the manuscript precise and easy to follow.

Nonetheless, I would like to suggest the following changes:

Line 14: I would suggest you just name it a new type of fruit wine made from red dragon fruit

Line 44: change ‘wine’ by fruit wine

Line 63: the same, change ‘red wine’ by fruit wine or similar

Line 87-88: Why the fermenters had so large headspace? Isn’t this a source of oxidation during fermentation. I suggest you change this in future experimental designs

Figure 2 ºBrix: how come there is still 8 ºBrix and wines are dry? Residual sugars are related to wine in fermentation density and 8 ºBrix correspond to 1.038 SG and still 4.1% potenial alcohol by volumen

Line 190-194:  and still lactic acid (ca. 2g/L) did not modify pH values at all? What is the total acidity values expressed as TH?

Table footnotes: all yeast species names shall be written in italics

Line 244: floral?

Line 282: ‘compared to’ instead of ‘than’

Line 287: different growth rate

Line 292: delbrueckii

Line 322: amino acid or vitamin deficiency

Line 329-330: at what concentration succinic acid can cause salty and bitter taste? Are those thresholds reached by these wines?

Line 380-383: are betacyanins pH sensitive? How can fruit wine pH values affect the nature of betacyanins?

Line 384-393: references do not follow MDPI style

Line 405-410: do betacyanins undergo stabilization processses producing shifts in maxl absorbance?

Line 411-420: all yeast species names in italics

Author Response

Responses to editor and reviewers’ comments

Reviewer 2 Comments:

Comment 1: Line 14: I would suggest you just name it a new type of fruit wine made from red dragon fruit

Response: As suggested, we have edited this in the revised manuscript.

Comment 2: Line 44: change ‘wine’ by fruit wine

Response: As suggested, we have edited this in the revised manuscript.

Comment 3: Line 63: the same, change ‘red wine’ by fruit wine or similar

Response: As suggested, we have edited this in the revised manuscript.

Comment 4: Line 87-88: Why the fermenters had so large headspace? Isn’t this a source of oxidation during fermentation? I suggest you change this in future experimental designs

Response: Ok, noted, thank you very much for your suggestion, we will increase the fermentation volume (300-400 mL) in our future work.

Comment 5: Figure 2 ºBrix: how come there is still 8 ºBrix and wines are dry? Residual sugars are related to wine in fermentation density and 8 ºBrix correspond to 1.038 SG and still 4.1% potenial alcohol by volume.

Response: ºBrix value represents the total soluble solid content based on refractive index. In the present study, solutes other than sugars such as organic acids, in dragon fruit juice also contribute to ºBrix value. The fruit wine is regarded as a dry wine because the total sugar residues is lower than 2 g/L (Table 1) in the final product.

Comment 6: Line 190-194: and still lactic acid (ca. 2 g/L) did not modify pH values at all? What is the total acidity values expressed as TA?

Response: Titratable acidity was not measured. The presence of various organic acids at different levels made the dragon fruit wine a buffer solution. The wine fermented with L. thermotolerans had higher level of lactic acid, but less malic acid compared to the other two treatments. L. thermotolerans and S. cerevisiae produced more acetic acid than T. delbrueckii; S. cerevisiae fermented juice was low in succinic acid. Overall the effect on pH could balance out each other.

Comment 7: Table footnotes: all yeast species names shall be written in italics

Response: As suggested, we have edited this in the revised manuscript.

Comment 8: Line 244: floral?

Response: Yes, it is floral, we have edited this in the revised manuscript.

Comment 9: Line 282: ‘compared to’ instead of ‘than’

Response: As suggested, we have edited this in the revised manuscript.

Comment 10: Line 287: different growth rate

Response: As suggested, we have edited this in the revised manuscript.

Comment 11: Line 292: delbrueckii

Response: As suggested, we have edited this in the revised manuscript.

Comment 12: Line 322: amino acid or vitamin deficiency

Response: As suggested, we have edited this in the revised manuscript.

Comment 13: Line 329-330: at what concentration succinic acid can cause salty and bitter taste? Are those thresholds reached by these wines?

Response: The threshold value of succinic acid was 0.1 g/L (Rotzoll et al., 2006), when the concentration of succinic acid is over 2.0 g/L, it could cause a salty and bitter taste. In general, the concentration of succinic acid in red wine ranges from 0.1 to 2.6 g/L, with the mean of 1.2 g/L (Coulter, Godden, Pretorius, 2004). The dragon fruit wine produced in this study (0.86-1.67 g/L) falls into the range of normal red wine, which is not supposed to cause negative effect on wine taste.

Coulter, A.; Godden, P. W.; Pretorius, I. 2004. Succinic acid-how is it formed, what is its effect on titratable acidity, and what factors influence its concentration in wine? Australian and New Zealand Wine Industry Journal19, 16-25.

Rotzoll, N., Dunkel, A., and Hofmann, T. 2006. Quantitative studies, taste reconstitution, and omission experiments on the key taste compounds in morel mushrooms (Morchella deliciosa Fr.). J. Agri. Food Chem. 54: 2705-2711.

Comment 14: Line 380-383: are betacyanins pH sensitive? How can fruit wine pH values affect the nature of betacyanins?

Response: Betacyanins are not pH sensitive. Betalains are in general stable over pH ranging from 3 to 7 (Azeredo, 2009). Betacyanins, which are glycosylated betalains, are more stable than aglycons. Therefore, the pH of the dragon fruit wine (pH 4) is not going to cause degradation of betacyanin. The decrease in betacyanins could be mainly due to the β-glucosidase activity of the yeasts that caused the hydrolysis of the glycosidic bonds in the glucosylated betacyanins and released the less stable betanidin (Esatbeyoglu et al., 2015).

Azeredo, H. M. C. 2009. Betalains: properties, sources, applications, and stability-a review. International Journal of Food Science & Technology, 44, 2365-2376.

Esatbeyoglu, T.; Wagner, A. E.; Schini-Kerth, V. B.; Rimbach, G. 2015, Betanin-A food colorant with biological activity. Mol. Nutr. Food Res. 59, 36-47.

Comment 15: Line 384-393: references do not follow MDPI style.

Response: As suggested, we have edited this in the revised manuscript.

Comment 16: Line 405-410: do betacyanins undergo stabilization processses producing shifts in maxl absorbance?

Response: Yes. Degradation of betacynin by enzyme or heat treatment caused color shift to yellow and orange hue which corresponds to lower wavelength; on the other hand, stabilization process shifted the maximum absorbance to upper wavelength (Stintzing, & Carle, 2004). The maximum absorbance of betacyanin is around 540 nm.

Stintzing, F. C.; Carle, R. 2004. Functional properties of anthocyanins and betalains in plants, food, and in human nutrition. Trends in Food Science & Technology, 15, 19-38.

Comment 17: Line 411-420: all yeast species names in italics

Response: As suggested, we have edited this in the revised manuscript.

Round 2

Reviewer 1 Report

I am satisfied with the changes. I understand that human tasting will belong to future work. However, I think that the levels of volatile compounds have to be compared to the ones usually found in wines, and indications of perception thresholds should be included in order to better understand the characteristics of this new product.

Author Response

Please find the responses in the attachment. Thank you.
